# Sacubitril/Valsartan Improves Diastolic Function But Not Skeletal Muscle Function in a Rat Model of HFpEF

**DOI:** 10.3390/ijms22073570

**Published:** 2021-03-30

**Authors:** Antje Schauer, Volker Adams, Antje Augstein, Anett Jannasch, Runa Draskowski, Virginia Kirchhoff, Keita Goto, Jeniffer Mittag, Roberta Galli, Anita Männel, Peggy Barthel, Axel Linke, Ephraim B. Winzer

**Affiliations:** 1Laboratory of Molecular and Experimental Cardiology, TU Dresden, Heart Center Dresden, 01307 Dresden, Germany; volker.adams@mailbox.tu-dresden.de (V.A.); antje.augstein@tu-dresden.de (A.A.); RUNa96@gmx.de (R.D.); V_Kirchhoff@gmx.de (V.K.); keita_goto@hotmail.co.jp (K.G.); anita.maennel@tu-dresden.de (A.M.); peggy.barthel@tu-dresden.de (P.B.); Axel.linke@tu-dresden.de (A.L.); ephraim.winzer@tu-dresden.de (E.B.W.); 2Department of Cardiac Surgery, Carl Gustav Carus Faculty of Medicine, Technische Universität Dresden, Heart Centre Dresden, Fetscherstrasse 76, 01307 Dresden, Germany; anett.jannasch@tu-dresden.de (A.J.); jeniffer.mittag@tu-dresden.de (J.M.); 3Clinical Sensoring and Monitoring, Department of Anesthesiology and Intensive Care Medicine, Faculty of Medicine, TU Dresden, 01307 Dresden, Germany; roberta.galli@tu-dresden.de

**Keywords:** HFpEF, ZSF-1, Sacubitril/Valsartan, diastolic dysfunction, skeletal muscle dysfunction

## Abstract

The angiotensin receptor/neprilysin inhibitor Sacubitril/Valsartan (Sac/Val) has been shown to be beneficial in patients suffering from heart failure with reduced ejection fraction (HFrEF). However, the impact of Sac/Val in patients presenting with heart failure with preserved ejection fraction (HFpEF) is not yet clearly resolved. The present study aimed to reveal the influence of the drug on the functionality of the myocardium, the skeletal muscle, and the vasculature in a rat model of HFpEF. Female obese ZSF-1 rats received Sac/Val as a daily oral gavage for 12 weeks. Left ventricle (LV) function was assessed every four weeks using echocardiography. Prior to organ removal, invasive hemodynamic measurements were performed in both ventricles. Vascular function of the carotid artery and skeletal muscle function were monitored. Sac/Val treatment reduced E/é ratios, left ventricular end diastolic pressure (LVEDP) and myocardial stiffness as well as myocardial fibrosis and heart weight compared to the obese control group. Sac/Val slightly improved endothelial function in the carotid artery but had no impact on skeletal muscle function. Our results demonstrate striking effects of Sac/Val on the myocardial structure and function in a rat model of HFpEF. While vasodilation was slightly improved, functionality of the skeletal muscle remained unaffected.

## 1. Introduction

Patients suffering from heart failure with preserved ejection fraction (HFpEF) take a large share of all heart failure (HF) patients and account, based on the underlying definition, for 22–73% of all cases [1], thereby predominantly affecting female patients and the elderly [2]. Common additional risk factors are obesity, hypertension, and diabetes [3]. Besides the complexity of diagnosis, the precise pathophysiological processes leading to HFpEF are incompletely understood, which is also related to the challenge of finding a suitable animal model for research [4]. One promising candidate is the ZSF1 (Zucker fatty and spontaneously hypertensive) obese rat [5]. It was developed by crossing rat strains with two separate leptin receptor mutations (fa and fa^cp^), the lean female ZDF rat (+/fa), and the lean male SHHF (spontaneously hypertensive heart failure) rat (+/fa^cp^). Offspring being homozygous for both mutations (fa:fa^cp^) are obese and develop insulin resistance, hyperglycemia, and mild hypertension (ZSF1-obese). The heterozygous offspring (ZSF1-lean) are lean and exhibit no signs of obesity and diabetes. In a previous study, we documented the development of HFpEF in adult ZSF1-obese animals based on diastolic dysfunction and clinical signs of chronic heart failure despite preserved left ventricular ejection fraction (LVEF) at an age of 20 weeks, suggesting the ZSF1 rat to serve as an ideal model for studying HFpEF and potential therapies [5].

While progress has been made in the treatment of HFrEF (heart failure with reduced ejection fraction, LVEF < 40%) [6], suitable treatment strategies for HFpEF are still lacking and previous efforts have failed to improve the outcome in HFpEF [7,8]. Approaches range from the treatment of comorbidities, i.e., by physical activity and caloric restriction [9], to conventional HF therapies, i.e., ACE inhibitors [10], AT1 antagonists [11], or beta-blockers [12], and innovative but still experimental approaches that intervene in underlying signaling pathways, like the nitrogen monoxide–cyclic guanosine monophosphate–protein kinase (NO–cGMP–PKG)-axis [13].

The angiotensin receptor neprilysin inhibitor LCZ696 (Sac/Val), which combines valsartan and sacubitril had been proven to be superior to enalapril in HFrEF patients regarding the combined primary endpoint of cardiovascular death and heart failure hospitalization (PARADIGM-HF trial) [6]. However, in the PARAGON-HF trial, which enrolled 4822 patients with HFpEF, Sac/Val failed to reduce the same composite endpoint in comparison to Valsartan [8,14]. Nevertheless, subgroup analyses of this trial suggest beneficial effects in patients with mildly reduced LVEF and females, suggesting a benefit of Sac/Val in specific subgroups [15]. Very recently, preliminary data from the PARALLAX-HF trial demonstrated superiority of Sac/Val compared to individualized medical therapy in the reduction of NT-proBNP levels associated with a lower clinical event rate [16]. Beyond a possible clinical impact of Sac/Val in HFpEF, the underlying pathophysiological mechanisms of Sac/Val treatment within the HFpEF syndrome are incompletely understood.

Therefore, the present study aimed to address the question, whether female ZSF1 obese rats with HFpEF phenotype benefit from a 12-week treatment with Sac/Val as a secondary prevention strategy, regarding myocardial, vascular, and skeletal muscle function.

## 2. Results

### 2.1. Baseline Animal Characteristics at 20 Weeks of Age

At an age of 20 weeks, obese animals presented with the indication of a pronounced HFpEF compared to their age-matched lean control group (Table 1). Obese ZSF1 animals were characterized by obesity, enhanced serum glucose levels, myocardial hypertrophy, and arterial hypertension. While the systolic function was preserved (77% in lean controls vs. 80% in obese rats), diastolic function was significantly impaired, as evaluated by the ratio of E/é (ZSF1 lean: 19.8 ± 0.5 vs. ZSF1 obese: 25.0 ± 0.7; *p* < 0.001), and left ventricular end diastolic pressure (LVEDP; 15 mmHg in lean controls vs. 20 mmHg in obese rats). A detailed baseline physiologic, echocardiographic, and hemodynamic evaluation of the 20-week-old animals is shown in Table 1.

Perivascular as well as overall LV fibrosis were analyzed and revealed significantly enhanced collagen levels in both the vessel environment and the left ventricular tissue (Figure 1A–C). Moreover, endothelial-dependent (Figure 1D) and independent (Figure 1E) vasodilating properties of carotid arteries were impaired in the ZSF1-obese animals.

Regarding the skeletal muscle we measured a significant lower weight of the tibialis anterior (TA, Figure 2A) in the obese group whereas the weight of Soleus (Figure 2B) and extensor digitorum longus (EDL, Figure 2C) was not significantly different between the groups. While absolute force of EDL and Soleus were comparable between both groups (Figure 2D,E), specific force was lower in the EDL (Figure 2F) but not in the Soleus (Figure 2G).

### 2.2. Animal Characteristics at 32 Weeks of Age (After 12-Week Sac/Val or Control)

At an age of 20 weeks, 15 lean and 30 obese rats were included in the 12-week lasting experimental phase with 15 obese animals randomly chosen for Sac/Val treatment and 15 obese and 15 lean animals serving as controls. In the Sac/Val group, we lost 2 animals following oral gavage in week 2 due to a bitten off plastic feeding tube. Table 2 lists physiologic, echocardiographic, and hemodynamic parameters of all three groups at an age of 32 weeks. In both the Sac/Val and the obese control group, animals presented with more than twice the body weight of the lean animals. However, the overall heart weight and specifically the LV weight were significantly reduced in the Sac/Val group compared to the obese control. Enhanced levels of serum glucose and HbA1c in obese control rats compared to their lean controls were not significantly reduced by Sac/Val treatment. However, HbA1c levels of the treated animals were not significantly higher than those of the lean control animals. Obese control rats presented with elevated kidney weights, which were even increased in the Sac/Val group.

### 2.3. Impact of Sac/Val on Systolic and Diastolic Function

In all groups, LVEF was preserved and ranged between 74% and 80% at 32 weeks of age (Table 2, Figure 3A). Considering the temporal development over the 12-week treatment period, the LVEF in the obese control group slightly decreased while it remained stable on a high level in the Sac/Val-group, a difference which became significant after 8 weeks. At the beginning of the experiment, the ratio E/é was similar in both obese groups (obese: 25.4; Sac/Val: 25.9) and significantly higher compared to the lean control (lean: 19.8; Table 3, Figure 3B). While the E/é-curves of the lean and the obese control group showed an almost parallel development over time, in the Sac/Val group, E/é declined more and more and first became significantly lower than its obese control after four weeks of treatment (Figure 3B). Representative ultrasound pictures of Pulsed Wave Doppler and Tissue Doppler measurement (Figure 3C) reveal that the elevated E/é ratio in the obese control group is the result of both increased early filling velocity and decreased early tissue velocity. In the Sac/Val group, both parameters became restored.

### 2.4. Impact of Sac/Val on Invasively Determined Hemodynamics

To examine and compare pressure/volume conditions at the experimental endpoint, invasive hemodynamic measurements were performed in the left and in the right ventricle as well as in the ascending aorta of all groups (Figure 4). While the LVESP and the LVEDP were significantly enhanced in the obese control compared to the lean animals (LVESP: 165 mmHg vs. 113 mmHg in the lean control; LVEDP: 22 mmHg vs. 15 mmHg in the lean control), Sac/Val treatment significantly reduced LVESP (137 mmHg) and LVEDP (16 mmHg) compared to the obese control (Figure 4E,F). Representative baseline PV loops visualize striking differences between the groups (Figure 4A). Transient occlusion of the inferior vena cava was performed to obtain load independent indexes of inotropy and diastolic compliance (Figure 4B–D). LVs of the obese control group were more contractile than LVs of the lean controls (Figure 4G) but also less compliant, regarding the increased stiffness constant β_W_ (Figure 4H). Compared to the obese group, inotropy was slightly decreased in the Sac/Val group. However, LV stiffness showed a clear trend to be reduced compared to the obese control group (Figure 4G,H).

Comparison of hemodynamic conditions in the right ventricle revealed a reduced volume capacity of Sac/Val treated animals vs. the obese controls, possibly going back to the smaller heart size in this group. The mean arterial pressure (MAP) in the ascending aorta was increased in obese animals (136 mmHg vs. 105 mmHg in the lean control group) but significantly reduced by Sac/Val treatment (120 mmHg, Table 2).

### 2.5. Impact of Sac/Val on Myocardial Fibrosis and Titin Phosphorylation

While perivascular fibrosis was enhanced in obese control rats vs. the lean control, it was significantly reduced in the Sac/Val treated rats compared to the obese control (Figure 5A,B). For the evaluation of overall LV fibrosis, the mRNA expression of Collagen Type I and -III (Col1a1, Col3a1) was measured in LV tissue and resulted in 35% higher Col1a1 expression and 43% increased Col3a1 expression in ZSF1-obese rats compared to the lean control (Figure 5C,D). Sac/Val treatment significantly reduced the expression of Col1a1 by 10% and the expression of Col3a1 by 26% compared to the obese control, respectively. 

Activity of the Matrix-Metalloproteinase-2 (MMP-2), a gelatinase involved in processes of myocardial remodeling, was increased by 25% in the obese control compared to lean rats (Figure 5E). Sac/Val treatment reduced the MMP-2 activity to less than one third of the obese control. In ZSF1-obese control rats, the LV expression of the natriuretic peptides ANP and BNP, indicators of myocardial stress, was elevated by 40% and 70%, respectively, compared to the lean control rats (Figure 5F,G). In Sac/Val treated animals, we measured 29% less ANP expression than in obese control rats while the BNP expression was even at the level of the lean control group. Although the plasma levels of NT-proBNP were slightly lower in the Sac/Val treated group, they did not differ significantly from the control obese group (Table 2). To gain insight into underlying mechanisms, which are possibly involved in the observed alterations, we analyzed the concentration of the second messenger cGMP and the phosphorylation status of titin in LV tissue of all groups (Figure 5H,I). While cGMP levels in the obese control group were slightly but insignificantly decreased compared to the lean control rats, they were enhanced by almost 80% in the Sac/Val treated animals compared to the obese control (Figure 5H), indicating a possible role for cGMP in the Sac/Val triggered signaling cascade. Titin, an important structural protein of the sarcomere, which is responsible for the passive muscle elasticity, has been reported to mediate stiffness if it is hypo-phosphorylated. In our setting, titin of obese control rats was significantly lower phosphorylated than titin of the lean control rats (Figure 5I). Sac/Val treated animals presented phospho-titin levels which were slightly higher than those measured in the lean control group and significantly elevated compared to the obese control.

### 2.6. Impact of Sac/Val on Fat Deposition within the Mitral Valve Annulus

To estimate the extent of intramyocardial fat deposition and its potential role in the observed diastolic dysfunction, native cryo-embedded heart slices were used for brightfield microscopy and nonlinear optical microscopy (NLOM) at the level of the mitral valve annulus. Representative pictures are shown in Figure 6. When we compared sections of all three groups, a high degree of fat deposition was obvious in obese control rats (Figure 6C,D), while lean control rats barely showed any fat inserts at all (Figure 6A,B). Mitral-valve annuli of Sac/Val treated obese rats presented with strikingly reduced fat deposits compared to the obese control (Figure 6E,F).

### 2.7. Impact of Sac/Val on Vascular Function

Vascular function was analyzed in the left carotid artery (Figure 7) and revealed impaired maximal tension in animals of the obese control group compared to the lean rats (Figure 7A). Contractility of Sac/Val treated animals did not show significant differences compared to the obese control. Compared to the lean control endothelial-dependent vasodilation was significantly lower in the obese control group (Figure 7B) and unchanged in Sac/Val treated animals (Figure 7C). Sac/Val treatment slightly improved endothelial-dependent vasodilating properties compared to the obese control group (Figure 7B).

### 2.8. Impact of Sac/Val on Skeletal Muscle Atrophy and Skeletal Muscle Function

Muscle weight/tibia length ratio and cross-sectional area (CSA) were compared between all three groups and revealed reduced values in both the obese control group and the obese Sac/Val group compared to the lean control (Figure 8A,B). In contrast, there was no difference between lean and obese control regarding muscle weight/tibia length ratio of both Soleus (Figure 8C) and EDL (Figure 8D). However, Sac/Val treated animals showed slightly increased values compared to the lean control.

While we did not observe any differences in EDL absolute force between all three groups (Figure 8E), lean control rats were superior in EDL specific force (Figure 8G) and in absolute as well as in specific force of the Soleus muscle (Figure 8F,H). Sac/Val did not differ from the obese control group in any of these approaches, suggesting no influence of the drug on skeletal muscle function.

## 3. Discussion

While huge progress has been made in the treatment of patients suffering from HFrEF [6], therapy of HFpEF remains challenging and a variety of approaches have failed to improve prognosis [7,8]. Sac/Val has been shown to exert beneficial effects at least in certain subgroups [8,14,15]. The present study aimed to examine the impact of a 12-week lasting treatment with Sac/Val on myocardial, vascular, and skeletal function in female ZSF1 obese rats with distinct signs of HFpEF. Our main findings are the following:(a)A 12-week treatment of obese ZSF1 rats with Sac/Val improved diastolic dysfunction, which became significant after four weeks of treatment and continued to improve until final assessment.(b)After 12 weeks of Sac/Val treatment, we observed significantly lower LV pressure in systole (LVESP) and diastole (LVEDP) as well as reduced left ventricular stiffness compared to the obese control group.(c)Sac/Val treatment significantly reduced left ventricular collagen expression levels and, in particular, perivascular fibrosis. Moreover, Sac/Val restored LV titin phosphorylation, suggesting both reduced fibrosis and normalized phosphorylation levels of titin to be key modules of the preserved left ventricular elasticity.(d)Although Sac/Val treatment did not affect body weight, we monitored reduced heart weight and decreased myocardial fat deposition. Reduced fat deposition at the site of the mitral valve annulus in the Sac/Val-treated group might indicate an enhanced mobility of this structure, possibly being involved in the improvement of diastolic function.(e)Sac/Val treatment slightly improved endothelial-dependent vasodilation in carotid arteries compared to the obese control group.(f)Sac/Val treated obese rats neither differed in skeletal muscle weight nor in skeletal muscle function compared to the obese control.

Taken together, the observed beneficial effects of Sac/Val on the myocardium suggests a potential use of the combinatory drug for the treatment of HFpEF.

### 3.1. The Female, Obese ZSF1 Rat as Research Model for HFpEF

Obese ZSF1 rats develop symptoms of HFpEF, already at a young age, and therefore have been demonstrated to be suitable for the research of HFpEF [5]. Severity and progression of HFpEF have been presented to be similar between male and female obese ZSF1 rats [17]. In the present study, female ZSF1 rats were chosen because HFpEF has been shown to predominantly affect elderly women [2] and because Sac/Val treatment has been proven to cause positive effects in a female subgroup of the PARAGON-HF trial [15]. Animals of all groups were included in the experiment at an age of 20 weeks. Subsets of 10 lean and 10 obese rats, which were randomly chosen for basic characterization, clearly confirmed relevant signs of HFpEF in obese animals (Table 1, Figure 1 and Figure 2) at the time point of treatment initiation. 

### 3.2. Impact of Sac/Val on the Myocardium

Over the course of the experiment, the ratio E/é in the Sac/Val treated group continuously declined and became first significantly lower than its obese control group after four weeks of treatment (Figure 3B). Moreover, 12 weeks of Sac/Val treatment improved the hemodynamic conditions in the left ventricle and the diastolic compliance, as indicated by reduced left ventricular stiffness (Figure 4). To estimate changes of the ventricular stiffness we determined LVEDP and the chamber stiffness constant β_W_, which are widely accepted to serve as load and chamber size-independent parameters of passive chamber properties [18,19]. Therefore, it is unlikely that the observed reduced LV stiffness in the treated group was a result of pure blood pressure reduction. Our findings were accompanied by reduced myocardial fibrosis (Figure 5A–D) and restored titin phosphorylation (Figure 5I), both indicating preserved left ventricular elasticity. The observed reduced left ventricular collagen expression levels in the Sac/Val treated group of the present study (Figure 5A–D) are in accordance with Sac/Val studies in patients, expressing reduced profibrotic biomarkers in both HFrEF 28 and HFpEF [20]. 

To date, there is only one comparable study evaluating the effects of Sac/Val on diastolic function in an experimental setting: Croteau et al. compared the effects of Valsartan (Val) alone and Sac/Val in C57BL/6J mice with obesity-related metabolic heart disease (MHD), which developed diastolic dysfunction, hypertrophy, and interstitial fibrosis [21]. Both drugs attenuated myocardial hypertrophy to a similar degree, whereas diastolic dysfunction and fibrosis were significantly reduced by Sac/Val but not by Val. This is in accordance with the findings of Suematsu et al., who reported Sac/Val but not Val to suppress myocardial fibrosis in a HFrEF mouse model with diabetes and ischemia reperfusion injury [22]. Therefore, both studies suggest that the superior effects of Sac/Val regarding myocardial fibrosis might be due to Neprilysin inhibition. 

However, the study of Croteau et al. bears some important limitations [23]. First of all, the study started up with 8-week-old, healthy C57BL/6J mice with no evidence of heart failure at the beginning of the treatment. Furthermore, treatment was initiated at the same time the MHD inducing diet was supplied, which is why the study describes the effects of a primary prevention rather than these of a secondary prevention. In our study, ZSF1 obese rats received Sac/Val treatment after the existence of HFpEF was confirmed. Therefore, to our knowledge, this is the first experimental study evaluating the effect of Sac/Val as a secondary preventive strategy for HFpEF. 

Nevertheless, the results of Croteau et al. give important insights into the different effects caused by both drugs and raise new hypotheses about the preventive spectrum of Neprilysin inhibition. Neprilysin inhibition has been reported to increase levels of circulating and tissue-bound natriuretic peptides [24], which in turn trigger the expression of cyclic guanosine monophosphate (cGMP), an important second messenger for cardiac function [25]. Its multitude of protective actions includes the activation of the protein kinase G (PKG), which has been shown to decrease myocardial stiffness via phosphorylation of titin [26]. Hamdani et al. have demonstrated that enhanced myocardial stiffness in obese ZSF1 rats is largely attributable to hypophosphorylation of titin [27], which is in accordance with the results of the present study (Figure 5I). 

We found reduced left ventricular expression levels of both ANP and BNP as well as lower plasma levels of NT-proBNP in the Sac/Val treated rats compared to the obese control group (Figure 5F,G, Table 2), which is in line with reduced myocardial wall stress as shown in the hemodynamic assessment. However, both the myocardial cGMP concentration and titin phosphorylation were significantly increased compared to the obese control group (Figure 5H,I) indicating an increased natriuretic peptide (NP) – NP receptor – cGMP signaling due to the diminished NP breakdown following Neprilysin inhibition. Other substrates of Neprilysin, like CNP [28], bradykinin, substance P, or adrenomedullin might contribute to these beneficial effects [29]. Our results suggest that decreased fibrotic processes and preserved titin phosphorylation might be the key targets of the observed improved diastolic function.

Although we measured similar body weights between the treated and untreated obese group, heart weights were significantly reduced in Sac/Val treated animals (Table 2) accompanied by a nonsignificant decrease in left ventricular wall thickness. This reduction in LV mass might be a consequence of blood pressure control and subsequent reduction in myocyte hypertrophy and interstitial fibrosis since we measured lower systolic and diastolic pressure in LV and aorta and reduced collagen expression within the LV in Sac/Val treated rats compared to control. On the other hand, Wu et al. demonstrated that patients suffering from HFpEF, especially women, had significantly more intramyocardial fat than HFrEF patients or non-HF controls and that the amount of intramyocardial fat correlated with LV diastolic dysfunction parameters [30]. We therefore determined intramyocardial fat deposition at the site of the mitral valve annulus in selected animals as demonstrated in Figure 6 and found more fat deposits in ZSF1-obese rats versus healthy controls. Surprisingly, fat deposition in Sac/Val treated animals was found to be reduced. In addition to these morphological changes, myocardial fat accumulation, particularly within the mitral valve annulus, might impact functional properties of contractility and relaxation. Lower fat content of the basal LV and mitral annulus potentially allows higher mobility of this structure and contributes to a lower E/é ratio. This concept needs to be further established to (I) quantify the effect size, (II) to analyze fat deposition at different sites within the heart muscle, (III) to address its pathophysiological impact, (IV) to address underlying pharmacological mechanisms. Interestingly, in HFrEF patients with chronic functional mitral regurgitation, the PRIME study revealed Sac/Val to be superior to Val regarding improved functional mitral regurgitation and reduced ratio of E/é [31]. These findings probably largely depend on the marked effects of Sac/Val on reverse LV remodeling in the dilated HFrEF ventricle. However, further studies will have to demonstrate if reduced fat accumulation within the mitral annulus or mitral apparatus following Sac/Val treatment contribute to improved function which depends on a balanced interaction of mitral annulus, leaflets, chordae tendineae, papillary muscles, and LV wall. 

### 3.3. Sac/Val Slightly Improves the Endothelial Function

Comorbidities going along with HFpEF, such as hypertension and diabetes mellitus, have been described to impair not only the myocardium but also central vessels as well as the microcirculation [32].

During the baseline characterization, vasodilating properties were measured in vessel rings of carotid arteries of 20-week-old lean and obese ZSF1 rats and revealed reduced endothelial-dependent (Figure 1D) and -independent (Figure 1E) vasodilation in the obese group. This is in accordance with a previous study of our group, demonstrating impaired endothelium-dependent vasodilation in the aorta of 20-week-old ZSF1 obese animals [33]. Our results also agree with the findings of Leite et al., who compared aortic ring preparations of 12-week-old ZSF1 lean and obese rats and healthy WKY rats and found decreased aortic compliance, impaired direct NO donor-mediated and endothelium-mediated vasodilation [34]. Moreover, Franssen et al. report microvascular inflammatory endothelial activation, oxidative stress, eNOS uncoupling, and impaired cGMP-PKG signaling in left ventricles both 20-week-old ZSF1 obese rats and HFpEF patients [35].

After 12 weeks of Sac/Val treatment, we only measured a small improvement of the endothelial-dependent vasodilation of carotid arteries compared to the obese control group (Figure 7), suggesting that Sac/Val exerts its main effects on the myocardium and not on the vasculature. However, further studies with focus on other vessel types will be necessary to confirm the present results.

### 3.4. Sac/Val Neither Influences Skeletal Muscle Atrophy nor Skeletal Muscle Function

An urgent clinical symptom of heart failure patients is exercise intolerance which leads not only to reduced quality of life but also to alterations in the peripheral skeletal muscle, including molecular changes like the switch of the fiber type, muscle atrophy, and mitochondrial energy production [36]. Whether exercise training is capable of improving these alterations is discussed controversially [36,37], and a pharmaceutical approach would be desirable.

We recently reported that the ZSF1 obese rat reliably mimics muscular changes seen in HFpEF patients [37] which is in accordance with the data of the 20- and 32-week-old ZSF1 obese rats in the present study (Figure 2 and Figure 8). Compared to their lean counterparts, muscle atrophy was especially evident in the TA of ZSF1 obese rats, indicated by reduced muscle weight and lower CSA. Twelve weeks of Sac/Val treatment did not improve the observed atrophy. Functional measurements of specific and absolute force of Soleus and EDL (Figure 8) did not reveal beneficial effects of the drug on the skeletal muscle. Therefore, we suggest that Sac/Val treatment does not influence the skeletal muscle, but improves diastolic dysfunction and, at least to a small degree, the endothelial function in ZSF1 obese rats. 

It is noteworthy to mention that the present study is subject to some limitations. First of all, all experiments were performed with female rats and no data have been collected with male animals, which is why gender-dependent differences cannot be excluded. Further experiments will have to test for effects of Sac/Val in male HFpEF models. Moreover, due to species-dependent differences, our results cannot be directly translated to clinical conditions and have to be critically assessed by further studies. The suggested molecular pathways are not completely proven to be involved in the beneficial effects of Sac/Val and further experiments will be necessary to complete our understanding about the drugs mechanism of action.

## 4. Materials and Methods

### 4.1. Animals

The present study was performed with female obese ZSF1/SHHF hybrid-HFpEF rats (obese) and their lean controls (lean). A total number of 25 ZSF1-lean and 40 ZSF1-obese rats was obtained from Charles River Laboratories. At an age of 20 weeks, 10 lean and 10 obese rats were randomly selected for baseline characterization. The remaining obese rats were randomly divided into two equal sized groups with *n* = 15 receiving a daily oral gavage of Sac/Val (60 mg/kg/day, LCZ696, Novartis, Nürnberg, Germany) over a time period of 12 weeks (Sac/Val) and *n* = 15 serving as sedentary obese control. The 15 lean rats served as sedentary lean control (Figure 9). 

All rats were housed under standard conditions and maintained on normal rat chow. Body weight and food consumption were monitored weekly and echocardiography was performed once a month for evaluation of the myocardial performance. At their respective experimental endpoint (20 weeks and 32 weeks of age), all animals underwent functional measurements of myocardium, endothelium, and skeletal muscles. Organ weights of heart, kidney, and skeletal muscles (soleus, tibialis anterior (TA), Extensor digitorum longus (EDL)) were determined. Samples of skeletal muscles (Soleus, TA, and EDL) and heart tissue were either preserved in paraformaldehyde (4%, PBS-buffered) for histological approaches or snap frozen in liquid nitrogen for molecular analyzes. All procedures were licensed (TVV 42/2018) and carried out according to the institutional Animal Care guidelines as regulated by the German Federal law governing animal welfare.

### 4.2. Echocardiography

Rats were anaesthetized by isoflurane (1.5–2%) and placed on a controlled warming pad to maintain a constant body temperature. Transthoracic echocardiography was performed using a Vevo 3100 system and a 21 MHz transducer (FUJIFILM VisualSonics Inc., Amsterdam, Netherlands) to assess cardiac structure and function. For systolic function, B-mode and M-mode of parasternal long and short axes at the papillary muscle level were measured. Diastolic function was assessed in the apical four-chamber view using pulse wave Doppler (for measurement of early (E) and atrial (A) waves of the mitral valve velocity) and tissue Doppler (for measurement of myocardial velocity é) at the level of the basal septal segment in the septal wall of the left ventricle. Functional parameters (e.g., LVEF and stroke volume) and ratios of E/é and E/A were obtained using the Vevo LAB 3.2.6 software.

### 4.3. Hemodynamic Measurements and Pressure-Volume Analysis

Prior to organ harvest, invasive hemodynamic measurements were performed in the ascending aorta and in both the left (LV) and the right ventricle (RV) of anaesthetized (ketamine (105 mg/kg) and xylazine (7 mg/kg)) but spontaneously breathing rats as recently described [5]. Pressure-volume loops were recorded under baseline conditions and during transient occlusion of the inferior *vena cava* by external compression of the vessel to obtain load independent indexes of contractility and chamber stiffness. The obtained end-systolic and end-diastolic pressure-volume relationships (ESPVR, EDPVR) were fitted to linear and exponential functions, respectively, with the slope E_es_ indicating contractility and the chamber stiffness constant β displaying the grade of diastolic compliance. To take account of the different heart sizes between the groups, the left ventricular wall volume (V_w_) was used as a normalization factor (β* V_w_ = β_w_) as reported before [18,38]. Data were recorded and analyzed with LabChart8 software (ADInstruments Ltd, Oxford, UK).

### 4.4. Skeletal Muscle Function

The right EDL and the left soleus were dissected and mounted vertically in a Krebs–Henseleit buffer-filled organ bath between a hook and force transducer as recently reported [5]. Briefly, in vitro muscle function was assessed by platinum electrodes stimulating the muscle with a supra-maximal current (700 mA, 500 ms strain duration, 0.25 ms pulse width) from a high-power bipolar stimulator (701C; Aurora Scientific Inc., Aurora, ON, Canada). The muscle bundle was set at an optimal length (Lo) equivalent to the maximal twitch force produced followed by a 15 min equilibration period. A force-frequency protocol was then performed at 1, 15, 30, 50, 80, 120, and 150 Hz, separated by 1 min rest intervals. Muscle length was measured using a digital micrometer and the muscle was subsequently detached, trimmed free from fat and tendon, blotted dry on filter paper, and weighed. Muscle force (N) was normalized to muscle cross-sectional area (CSA, cm^2^) by dividing muscle mass (g) by the product of optimal length (cm) and estimated muscle density (1.06 g/cm^3^), which allowed specific force (in N/cm^2^) to be calculated. For CSA analyses, paraffin-embedded skeletal muscle sections (3 μm) were H&E-stained (haematoxylin and eosin, Sigma, Taufkirchen, Germany) according to the manufacturer’s protocol and CSA was determined using Imaging software Zen 3.0 (Carl Zeiss Microscopy GmbH, Jena, Germany). 

### 4.5. Assessment of Carotid Artery Function

Vascular function of carotid artery rings was analyzed in vitro (DMT multi myograph System 620M), where vessel rings (2–3 mm) were mounted between a hook and a force transducer in a buffer-filled organ bath (in mmol/L: 118 NaCl, 25 NaHCO_3_, 4.7 KCl, 1.2 KH2PO_4_, 1.2 MgSO_4_, 2.5 CaCl_2_, 5.5 glucose). Optimal preconstriction was determined by measuring force responses after stimulation with KCl and phenylephrine in preliminary experiments. Following an equilibration period of at least 10 min vessel segments were preconstricted to a transmural pressure of 10 kPa (75 mm Hg) and equilibrated for a further 30 min. Maximal constriction was assessed by adding potassium chloride (KCl, final concentration 100 mmol/L) to the buffer. After rinsing the ring carefully several times, phenylephrine (PE)-dependent constriction was determined by adding PE up to a concentration of 1 × 10^−6^ mol/L. Relaxation measurements were done after constriction with PE (3 × 10^−7^ mol/L) with increasing concentrations of acethylcholine (ACH; Sigma, Taufkirchen, Germany; 1 × 10^−9^ to 5 × 10^−6^ mol/L) and sodium nitroprusside (SNP; Sigma; 1 × 10^−10^ to 5 × 10^−6^ mol/L. Vessel tensions were normalized to vessel length, set to 100%, and vasodilations were calculated as percentages

### 4.6. Quantification of Hba1c, NT-proBNP and cGMP

The plasma concentration of HbA1c and NT-proBNP as well as the concentration of cGMP in LV tissue were determined using enzyme-linked immunosorbent assays (HbA1c and NT-proBNP: MBS, BIOZOL, Germany and cGMP: CUSABIO, Wuhan, China) according to the manufacturer’s protocols.

### 4.7. Immunohistochemistry

For measurement of perivascular collagen deposition, paraffin-embedded heart sections (3 μm) were stained with Picrosirius red and perivascular fibrosis around arteries, expressed as perivascular fibrosis ratio (PFR) was quantified as described by Dai and colleagues [39]. PFR was defined as the area of perivascular fibrosis divided by the area of the vascular wall, averaged over all quantifiable images of arteries taken from a section. 

In preparation for nonlinear optical microscopy (NLOM), a nondestructive and label-free technique based on multiphoton processes [40] freshly harvested hearts were fixed in 4% paraformaldehyde overnight followed by a 24-h-incubation in sucrose solution (20%) before cryo-embedding (Tissue-Tek^®^ O.C.T.™ compound, Sakura Finetek Germany GmbH, Umkirch, Germany). Cryo-embedded heart sections (8 µm) were placed on glass slides, kept moist by a drop of phosphate buffered saline, and natively used for NLOM which was performed using laser excitation provided by two picosecond near-infrared Erbium fiber sources. Technical details were previously described [41]. Three signals were simultaneously acquired and combined in one RGB image: coherent anti-Stokes Raman scattering (CARS) in the red channel, two-photon excited fluorescence (TPEF) in the green channel and second harmonic generation (SHG) in the blue channel. For the identification of lipid structures, we applied CARS [42] and SHG for the detection of highly ordered structures lacking of inversion symmetry like fibrillary collagen and the combination of SHG and TPEF for myosin [43,44]. To ensure comparability to the sonographic results of tissue Doppler velocity and due to NLOM being a time and computationally intensive method, we focused exclusively on the area of the septal mitral valve annulus.

### 4.8. Assessment of Titin Phosphorylation

Homogenized myocardial samples (20 µg) were separated using a 2% agarose gel as described elsewhere [45]. After O/N fixation (50% methanol/10% acid), gels were stained with Pro-Q Diamond for 1h and subsequently with SYPRO Ruby overnight. Phosphorylation signals for titin proteins on Pro-Q Diamond-stained gels were normalized to SYPRO Ruby-stained total protein signals. 

### 4.9. Zymography

Matrix metalloproteinase-2 (MMP-2) -activity was measured by gelatine zymography. LV tissue was homogenized in RIPA buffer (20-fold of wet weight) containing protease inhibitors (Inhibitor mix M, SERVA Electrophoresis GmbH, Heidelberg, Germany). Samples were subsequently centrifuged at 16,000× *g* for 5 min, with the supernatant isolated and protein content determined (BCA assay; Pierce). Equal amounts of protein (10 µg) were mixed with sample buffer (250 mmol/l Tris·HCl pH 7.4, 10% sodium dodecylsulfate, 20% glycerol, and 0.005% bromphenolblue) under nondenaturating conditions. After electrophoresis (10% polyacrylamide gel containing 1% gelatine), gels were washed for 1.5 h in renaturation buffer (2.5% Triton X-100) and incubated in incubation buffer (in mmol/l: 50 Tris pH 7.6, 10 CaCl_2_, 50 NaCl, and 0.05% Brij-35) at 37 °C for 72 h. Gels were then stained with 0.25% Coomassie Brilliant Blue R-250, and MMP activity was observed as cleared unstained regions. Densitometry was used to quantify activity (OneD scan, Vilber, France).

### 4.10. RNA Isolation and Quantitative Real-Time PCR

Total RNA was isolated from LV tissue using Qiazol reagent and miRNeasy Mini Kit (Qiagen, Hilden, Germany) following the standard protocols. cDNA was synthesized with the Revert AID™ H.

Minus First Strand Synthesis Kit (Thermo Scientific, Braunschweig, Germany) using oligo-dT primers. Real-time PCR was performed using the CFX384TM Real Time PCR System (Bio-Rad Laboratories GmbH, Feldkirchen, Germany) and Maxima SYBR Green qPCR Kit (Thermo Scientific, Braunschweig, Germany). PCR program for all primer sets (Table 1) was as follows: 95 °C for 8 min prior to 40 amplification cycles, each consisting of 95 °C for 10 s, 58 °C for 15 s, and 72 °C for 30 s, with a final extension step at 72 °C for 2 min. Melting point analysis was done to prove the identity of the PCR products. Relative quantification of gene expression was calculated by ΔΔCT method with Polr2a and Rpl-32 as housekeeping genes using BioRad CFX Manager Software (Bio-Rad Laboratories GmbH, Feldkirchen, Germany). The expression of specific genes was normalized to its expression in ZSF1-lean animals. Specific primer sequences are listed in Table 3.

### 4.11. Statistical Analyses

All data are expressed as mean ± SEM (standard error of the mean). For comparison of lean and obese animals within the baseline analyses, 2-sided Student’s *t*-test of equal variances was performed. One-way analysis of variance (ANOVA) followed by Bonferroni post hoc was used to compare all three groups of the treatment study. Two-way repeated measures ANOVA followed by Bonferroni post hoc was used to compare echo data between the groups over time and to assess contractile function. *p*-values below 0.05 were considered to be statistically significant.

## 5. Conclusions

With the present study we demonstrate that a Sac/Val treatment over a time period of 12 weeks exerts striking effects on the myocardial structure and function in a rat model of HFpEF. While the drug does not seem to affect the skeletal muscle physiology and function, we demonstrate slightly improved endothelial-dependent vasodilative properties.

## Figures and Tables

**Figure 1 ijms-22-03570-f001:**
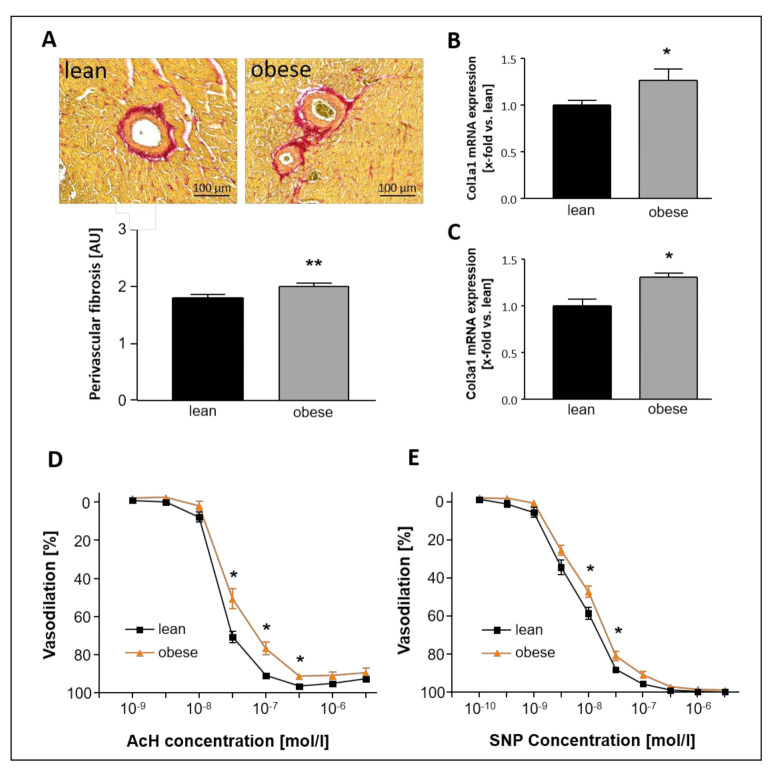
Baseline characterization of 20-week-old lean and obese Zucker fatty and spontaneously hypertensive (ZSF1) rats—myocardial fibrosis and vasodilative properties. (**A**) Brightfield microscopy of perivascular fibrosis in representative picrosirius-stained heart sections. (**B**,**C**) mRNA expression of Col1a1 (**B**) and Col3a1 (**C**) in LV tissue. (**D**) Endothelial-dependent vasodilation of sections of left carotid arteries in response to increasing concentrations of acetylcholine (AcH). (**E**) Endothelial-independent vasodilation of sections of left carotid arteries in response to increasing concentrations of sodium nitroprusside (SNP). Myocardial fibrosis: *n* = 10 for each group. Vascular function: *n* = 15 (lean and obese, respectively) and *n* = 13 (Sac/Val). Lean- and obese animals were compared by 2-sided Student’s *t*-test of equal variances. * *p* < 0.05, ** *p* < 0.01 vs. lean, respectively.

**Figure 2 ijms-22-03570-f002:**
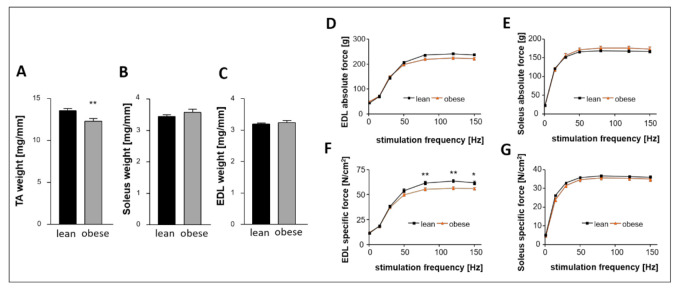
Basic characterization of 20-week-old lean and obese ZSF1 rats—skeletal muscle weight and function. Muscle weight of (**A**) TA muscle, (**B**) Soleus muscle, and (**C**) EDL muscle. (**D**,**E**) Absolute force measured for EDL (**D**) and Soleus (**E**) and specific force measured in EDL (**F**) and Soleus (**G**). *n* = 10 for each group. Lean and obese animals were compared by 2-sided Student’s *t*-test of equal variances. * *p* < 0.05, ** *p* < 0.01 vs. lean, respectively.

**Figure 3 ijms-22-03570-f003:**
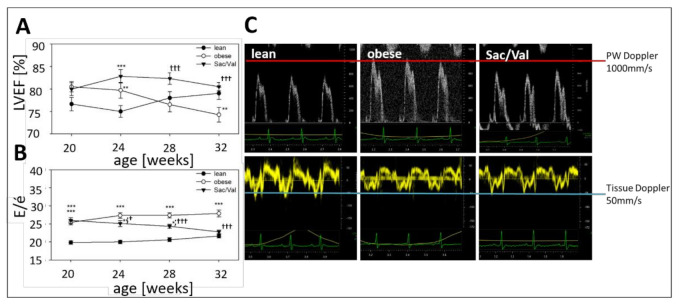
Development of cardiac function during 12 weeks of Sac/Val treatment. Systolic (**A**) and diastolic function (**B**) of lean, obese, and Sac/Val treated obese ZSF1 rats indicated by the LVEF and the ratio E/é. Comparison of representative pulsed wave (PW) Doppler and Tissue Doppler sequences after 12 weeks of Sac/Val or sedentary life style (**C**). *n* = 15 (each, lean and obese) and *n* = 13–15 (Sac/Val). To test the effect of Sac/Val two-way one-way analysis of variance (ANOVA) with repeated measures followed by Bonferroni post hoc was used to compare echo data between the groups over time. * *p* < 0.05, ** *p* < 0.01, *** *p* < 0.001 vs. lean, respectively. Ɨ *p* < 0.05, ƗƗƗ *p* < 0.001 vs. obese.

**Figure 4 ijms-22-03570-f004:**
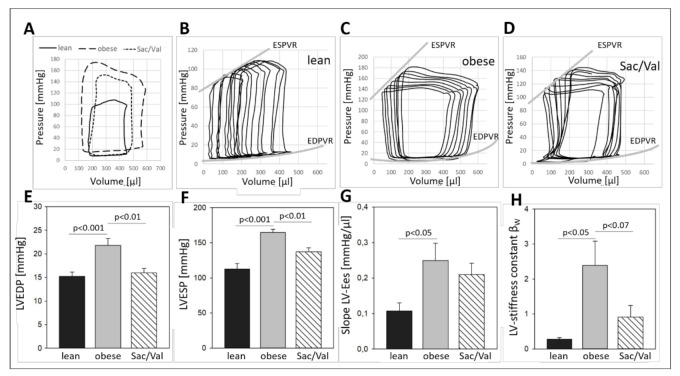
LV hemodynamic conditions and compliance after 12 weeks of Sac/Val treatment. Representative baseline Pressure-Volume (PV) loops of lean, obese, and Sac/Val treated rats at an age of 32 weeks (**A**). Representative PV loop shifts in response to a transient occlusion of the inferior vena cava of a lean (**B**), an obese (**C**), and a Sac/Val treated obese rat (**D**). (**E**) Comparison of LV end diastolic pressure (LVEDP) and (**F**) LV end systolic pressure (LVESP) at baseline conditions. Comparison of the LV contractility indicated by the slope LV-Ees obtained by the end-systolic pressure-volume relationships (ESPVR) (**G**). Comparison of the LV chamber stiffness indicated by the LV stiffness constant β_W_ (**H**) obtained by the end-diastolic pressure-volume relationships (EDPVR). *n* = 10–12. To test the effect of Sac/Val, one-way analysis of variance (ANOVA) followed by Bonferroni post hoc was used to compare groups.

**Figure 5 ijms-22-03570-f005:**
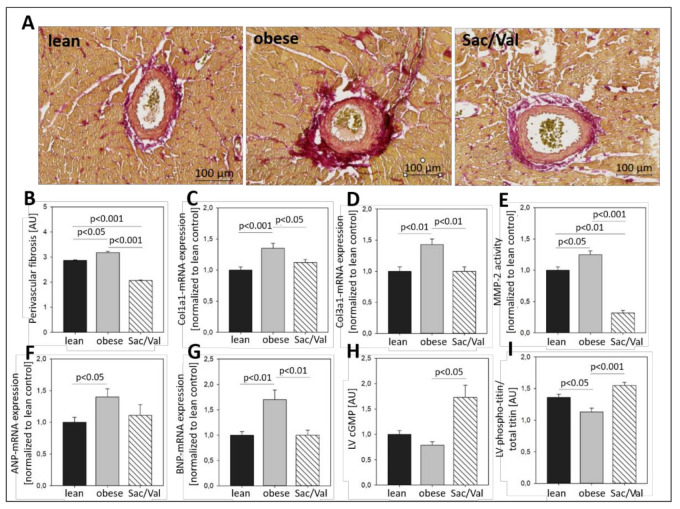
LV fibrosis and remodeling after 12 weeks of Sac/Val treatment. Brightfield microscopy of perivascular fibrosis in representative picrosirius-stained heart sections (**A**). Quantification of LV perivascular fibrosis (**B**). LV mRNA expression of Col1a1 (**C**) and Col3a1 (**D**). MMP-2 activity measured in LV tissue (**E**). LV mRNA expression of ANP (**F**) and BNP (**G**). cGMP concentration measured in LV tissue (**H**). LV protein expression of phosphorylated titin (**I**). *n* = 13–14. To test the effect of Sac/Val one-way analysis of variance (ANOVA) followed by Bonferroni post hoc was used to compare groups.

**Figure 6 ijms-22-03570-f006:**
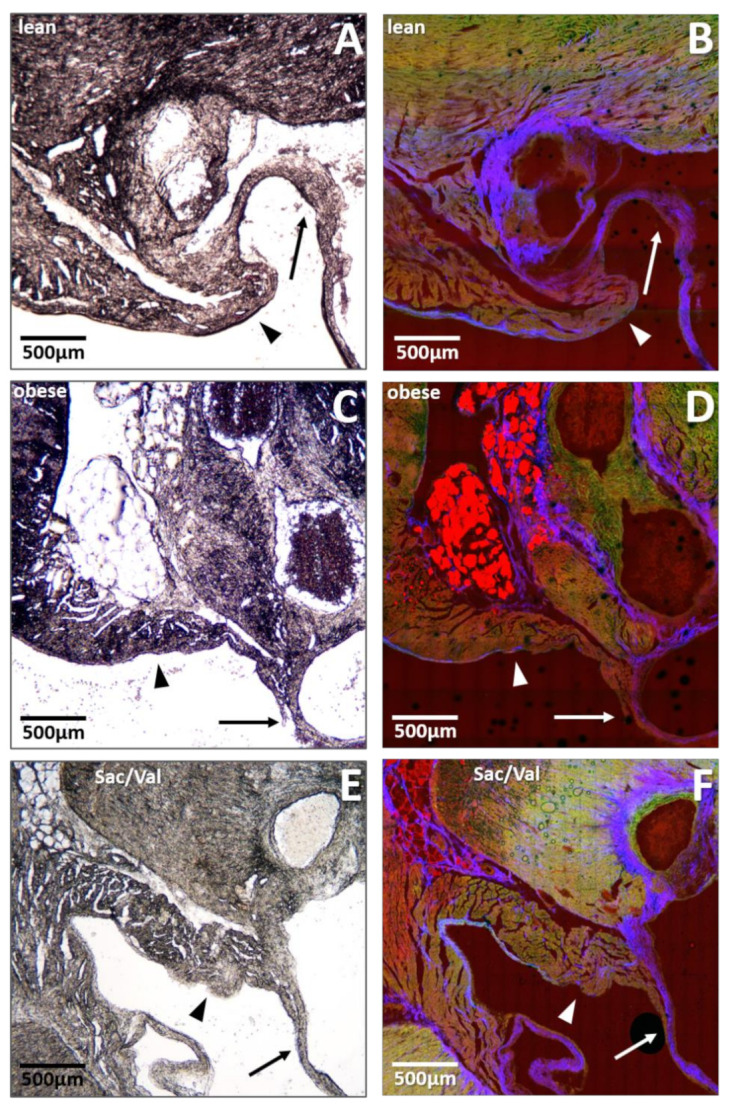
Comparison of LV fat deposits at the site of the mitral valve annulus after 12 weeks of Sac/Val. Brightfield microscopy (**A**,**C**,**E**) and nonlinear optical microscopy (NLOM) (**B**,**D**,**F**) of representative heart sections of a lean (**A**,**B**), an obese (**C**,**D**), and a Sac/Val treated obese rat (**E**,**F**). Arrows point to mitral valve structures. Arrow heads indicate the mitral valve annulus. NLOM visualizes collagen fibers (appearing in blue), myofilaments (appearing in green), and lipid structures (appearing in red).

**Figure 7 ijms-22-03570-f007:**
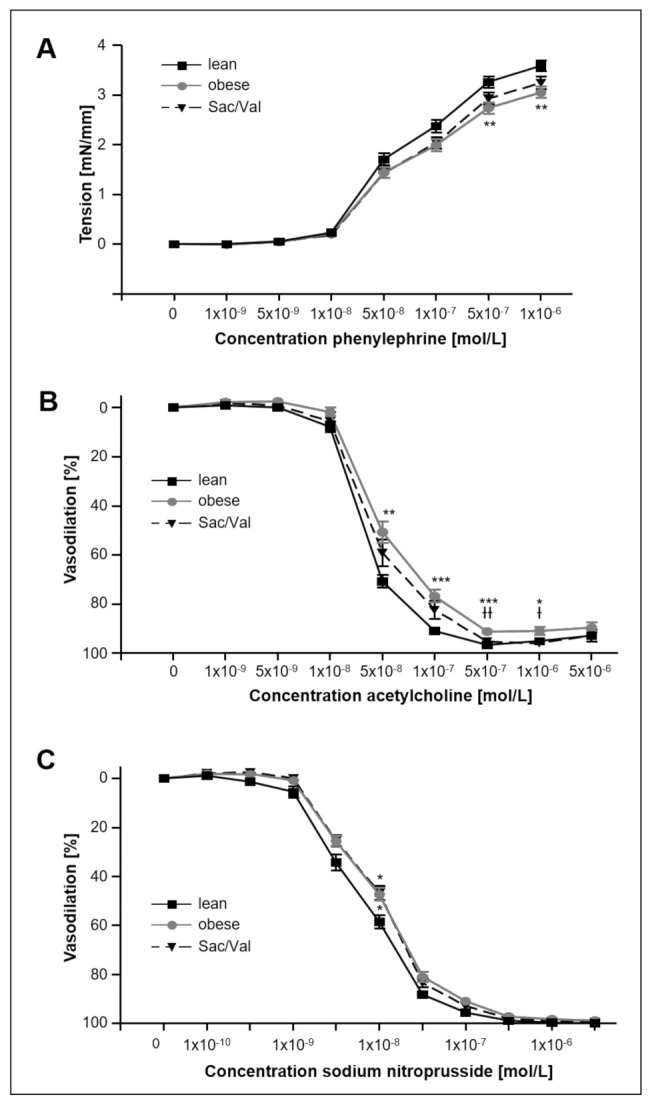
Contractile and vasodilative properties of left arterial carotid sections after 12 weeks of Sac/Val. Comparison of tension (**A**), endothelial-dependent (**B**), and endothelial-independent (**C**) vasodilation of left arterial carotid sections of lean, obese and Sac/Val treated obese ZSF1 rats after 12 weeks of Sac/Val or sedentary lifestyle. *n* = 13–15. To test the effect of Sac/Val, two-way ANOVA with repeated measures followed by Bonferroni post hoc was performed. * *p* < 0.05, ** *p* < 0.01, *** *p* < 0.001 vs. lean. Ɨ *p* < 0.05, ƗƗ *p* < 0.01 vs. obese.

**Figure 8 ijms-22-03570-f008:**
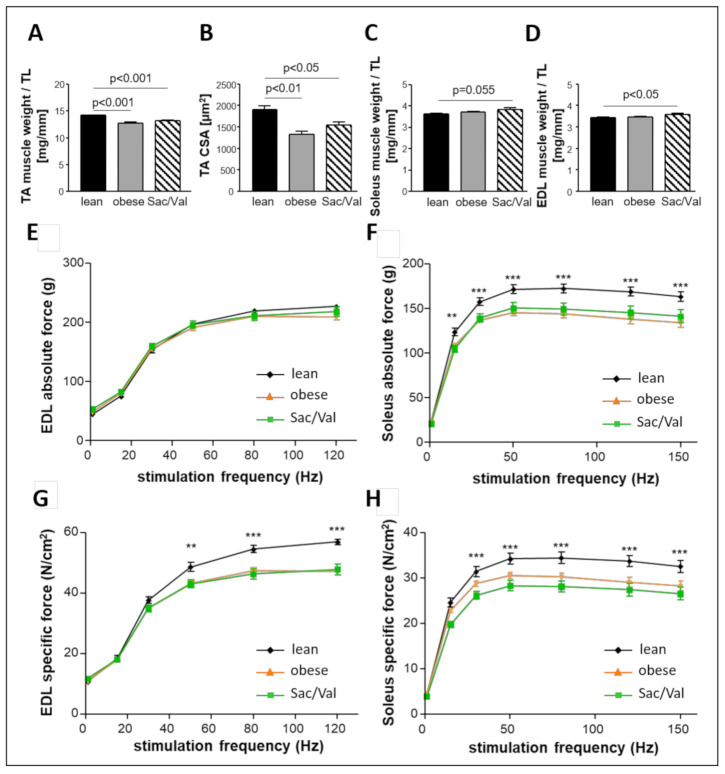
Skeletal muscle weight and function after 12 weeks of Sac/Val. Muscle weight (normalized to tibia length (TL)) (**A**) and cross-sectional area (CSA) of the TA muscle (**B**). Muscle weight of the Soleus (**C**) and the EDL (**D**), each normalized to tibia length. Absolute (**E**,**F**) and specific force (**G**,**H**) of the EDL (**E**,**G**) and Soleus (**F**,**H**). *n* = 13–15. To test the effect of Sac/Val, two-way ANOVA with repeated measures followed by Bonferroni post hoc was performed. ** *p* < 0.01, *** *p* < 0.001 vs. lean, respectively.

**Figure 9 ijms-22-03570-f009:**
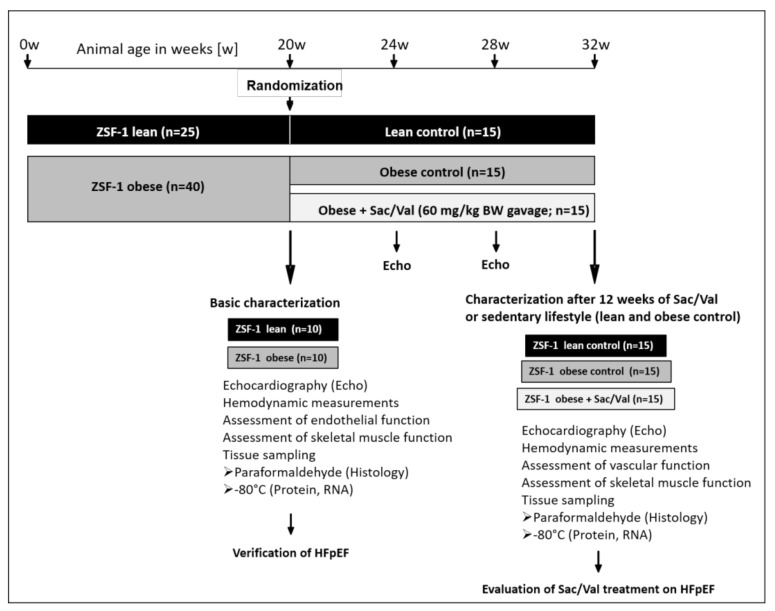
Study design. In the present study, a total number of 25 ZSF1-lean and 40 ZSF1-obese rats was included. At an age of 20 weeks, 10 lean and 10 obese rats were randomly selected for baseline characterization. The remaining obese rats were randomly divided into two equal sized groups with *n* = 15 receiving a daily oral gavage of Sac/Val (60 mg/kg/day, LCZ696, Novartis, Germany) over a time period of 12 weeks (Sac/Val) and *n* = 15 serving as sedentary obese control. The 15 lean rats served as sedentary lean control.

**Table 1 ijms-22-03570-t001:** Baseline characterization of the 20-week-old lean and obese animals.

Physiology	Lean (*n* = 10)	Obese (*n* = 10)	*p*-Value
Body weight (g)	239 ± 4	474 ± 8	**<0.001**
Tibia length (TL, mm)	36.25 ± 0.19	36.62 ± 0.12	0.8
Heart weight/TL (mg/mm)	23.05 ± 0.31	32.08 ± 0.34	**<0.001**
Kidney weight/TL (mg/mm)	24.85 ± 0.65	41.34 ± 1.08	**<0.001**
Serum glucose (mg/dL)	275 ± 14	444 ± 35	**<0.001**
Serum NT-proBNP (pg/mL)	42.67 ± 10.13	99.94 ± 22.62	0.03
**Echocardiography**			
Left ventricle (LV) weight (mg)	721 ± 21	1009 ± 28	**<0.001**
LVEF (%)	77 ± 2	80 ± 1	0.06
LVFS (%)	57 ± 2	60 ± 2	0.32
LVESV (µL)	339 ± 12	422 ± 18	**<0.001**
LVEDV (µL)	444 ± 18	525 ± 21	**<0.01**
E/é	19.8 ± 0.5	25.0 ± 0.7	**<0.001**
E/A	1.3 ± 0.04	1.5 ± 0.1	0.07
LAA (mm^2^)	11.4 ± 0.6	11.3 ± 0.6	0.6
LVAW (mm)	1.8 ± 0.1	2.2 ± 0.1	**<0.001**
LVPW (mm)	1.6 ± 0.1	2.0 ± 0.1	**<0.001**
LVEDD (mm)	6.4 ± 0.2	7.1 ± 0.1	0.06
**Invasive hemodynamics**			
*Ascending aorta*			
SAP (mmHg)	116 ± 5	152 ± 8	**<0.01**
DAP (mmHg)	86 ± 4	98 ± 6	0.1
MAP (mmHg)	99 ± 6	116 ± 5	0.07
*Left ventricle*			
Heart frequency [bpm)	238 ± 7	217 ± 8	0.1
LVEDP (mmHg)	15 ± 2	20 ± 1	**<0.05**
LVESP (mmHg)	108 ± 7	146 ± 6	**<0.01**
LVEDV (µL)	367 ± 25	441 ± 13	**<0.01**
LVESV (µL)	106 ± 16	136 ± 16	0.2
dP/dt max (mmHg/s)	5986 ± 378	9421 ± 914	**<0.001**
dP/dt min (mmHg/s)	−5588 ± 181	−7361 ± 183	**<0.001**
dV/dt max (µL/s)	11,404 ± 1044	10,505 ± 1478	0.7
dV/dt min (µL/s)	−8424 ± 569	−10,168 ± 1154	0.3
Tau (ms)	18 ± 1	17 ± 1	0.7
*Right ventricle*			
RVEDP (mmHg)	16 ± 1	17 ± 5	0.7
RVESP (mmHg)	42 ± 3	46 ± 5	0.5
RVEDV (µL)	108 ± 24	173 ± 28	**<0.05**
RVESV (µL)	79 ± 19	120 ± 17	**<0.05**
dP/dt max (mmHg/s)	1680 ± 231	2285 ± 118	0.08
dP/dt min (mmHg/s)	−1250 ± 159	−2355 ± 471	**<0.05**
dV/dt max (µL/s)	909 ± 76	1297 ± 346	0.2
dV/dt min (µL/s)	−1009 ± 152	−1192 ± 233	0.5
Tau (ms)	38 ± 5	34 ± 9	0.5

Values are presented as Mean ± SEM. LVEF: left ventricular ejection fraction; LVFS: left ventricular fractional shortening; LVEDP left ventricular end-diastolic pressure; LVESP: left ventricular end-systolic pressure; LVEDV: left ventricular end-diastolic volume; LVESV: left ventricular end-systolic volume; LAA: left atrial area; LVAW: left ventricular anterior wall; LVPW: left ventricular posterior wall; LVEDD: left ventricular end-diastolic diameter; SAP: systolic arterial blood pressure; DAP: diastolic arterial blood pressure; MAP: mean arterial blood pressure; RVEDP: right ventricular end-diastolic pressure; RVESP: right ventricular end-systolic pressure; RVEDV: right ventricular end-diastolic volume; RVESV: right ventricular end-systolic volume.

**Table 2 ijms-22-03570-t002:** Characterization of the 32-week-old lean and obese animals.

Physiology	Lean (*n* = 15)	Obese (*n* = 15)	Sac/Val (*n* = 13)
Body weight [g)	265 ± 4	559 ± 9 ^***^	563 ± 8 ^***^
Tibia length (TL, mm)	38.2 ± 0.1	38.0 ± 0.1	37.6 ± 0.2
Heart weight/TL (mg/mm)	23.54 ± 0.31	35.44 ± 0.50 ^***^	33.22 ± 0.42 ^***,^^ƗƗ^
Kidney weight/TL (mg/mm)	23.96 ± 0.45	43.21 ± 1.01 ^***^	47.51 ± 1.03 ^***,^ ^ƗƗ^
Serum glucose (mg/dl)	324 ± 20	571 ± 21 ^***^	577 ± 31 ^***^
Serum HbA1c (ng/ml)	4.69 ± 0.43	7.35 ± 0.95 ^*^	5.95 ± 0.70
Serum NT-proBNP (pg/ml)	96.96 ± 9.42	209.04 ± 38.26 ^*^	154.21 ± 28.12
**Echocardiography**			
LV weight (mg)	890 ± 11	1274 ± 20 ^***^	1070 ± 13 ^***,^^ƗƗƗ^
LVEF (%)	79.0 ± 1.3	74.3 ± 1.2 ^**^	80.5 ± 0.9 ^ƗƗƗ^
LVFS (%)	52 ± 1.7	54 ± 1.3	53 ± 1.4
LVESV (µL)	441 ± 28	511 ± 17 ^*^	469 ± 16
LVEDV (µL)	555 ± 34	691 ± 25 ^***^	585 ± 17 ^ƗƗƗ^
E/é	21.6 ± 0.5	27.9 ± 0.7 ^***^	22.8 ± 0.5 ^ƗƗƗ^
E/A	1.2 ± 0.1	1.4 ± 0.1 ^**^	1.4 ± 0.1
LAA (mm^2^)	17 ± 1.1	21 ± 0.8 ^**^	19 ± 1.2
LVAW (mm)	1.76 ± 0.06	2.03 ± 0.07 ^**^	1.86 ± 0.05
LVPW (mm)	1.66 ± 0.08	2.02 ± 0.09 ^**^	1.95 ± 0.07 ^*^
LVEDD (mm)	6.67 ± 0.11	7.97 ± 0.19 ^***^	7.96 ± 0.21 ^***^
**Invasive Hemodynamics**	*n* = 12	*n* = 11	*n* = 9
*Ascending aorta*			
SAP (mmHg)	125 ± 6	180 ± 4 ^***^	154 ± 4 ^**,^ ^ƗƗƗ^
DAP (mmHg)	95 ± 5	113 ± 3 ^**^	102 ± 3 ^ƗƗ^
MAP (mmHg)	105 ± 6	136 ± 3 ^***^	120 ± 3 ^*,^ ^ƗƗ^
*Left ventricle*			
Heart frequency (bpm)	226 ± 6	210 ± 14	222 ± 9
LVEDP (mmHg)	15 ± 1	22 ± 1^***^	16 ± 1 ^ƗƗ^
LVESP (mmHg)	113 ± 8	165 ± 5 ^***^	137 ± 10 ^ƗƗ^
LVEDV (µL)	434 ± 29	622 ± 47^**^	475 ± 27 ^Ɨ^
LVESV (µL)	134 ± 17	291 ± 31 ^***^	172 ± 18 ^ƗƗ^
dP/dt max (mmHg/s)	5791 ± 310	9864 ± 411 ^***^	10,042 ± 412 ^***^
dP/dt min (mmHg/s)	−6055 ± 411	−8148 ± 323 ^***^	−7549 ± 483 ^*^
dV/dt max (µL/s)	14,972 ± 1752	15,263 ± 926	12,636 ± 1622
dV/dt min (µL/s)	−12,495 ± 1344	−13,666 ± 1874	−11,082 ± 887
Tau (ms)	17 ± 1	17 ± 1	17 ± 1
*Right ventricle*			
RVEDP (mmHg)	15 ± 0.7	16 ± 0.8	18 ± 1.8
RVESP (mmHg)	35 ± 0.9	44 ± 1.3 ^***^	45 ± 3.4 ^**^
RVEDV (µL)	153 ± 15	314 ± 26 ^***^	204 ± 17 ^ƗƗ^
RVESV (µL)	113 ± 14	200 ± 24 ^**^	161 ± 17 ^*^
dP/dt max (mmHg/s)	1193 ± 23	2123 ± 81 ^***^	2293 ± 79 ^***^
dP/dt min (mmHg/s)	−1008 ± 45	−1248 ± 94 ^*^	−1159 ± 84
dV/dt max (µL/s)	1441 ± 157	2285 ± 198 ^**^	1103 ± 137 ^ƗƗƗ^
dV/dt min (µL/s)	−1283 ± 153	−2331 ± 197 ^***^	−1182 ± 127 ^ƗƗƗ^
Tau (ms)	40 ± 2	39 ± 3	40 ± 4

Values are presented as Mean ± SEM. * *p* < 0.05, ** *p* < 0.01, *** *p* < 0.001 vs. Lean, respectively. ^Ɨ^
*p* < 0.05, ^ƗƗ^
*p* < 0.01, ^ƗƗƗ^
*p* < 0.001 vs. obese, respectively. LVEF: left ventricular ejection fraction; LVFS: left ventricular fractional shortening; LVEDP left ventricular end-diastolic pressure; LVESP: left ventricular end-systolic pressure; LVEDV: left ventricular end-diastolic volume; LVESV: left ventricular end-systolic volume; LAA: left atrial area; LVAW: left ventricular anterior wall; LVPW: left ventricular posterior wall; LVEDD: left ventricular end-diastolic diameter; SAP: systolic arterial blood pressure; DAP: diastolic arterial blood pressure; MAP: mean arterial blood pressure; RVEDP: right ventricular end-diastolic pressure; RVESP: right ventricular end-systolic pressure; RVEDV: right ventricular end-diastolic volume; RVESV: right ventricular end-systolic volume.

**Table 3 ijms-22-03570-t003:** List of primers for qRT-PCR.

Gene	Primer 1	Primer 2	Gene bank ID
**ANP**	TCCCGTATACAGTGCGGTGTC	GGAGGCATGACCTCATCTTC	NM_012612
**BNP**	ACAATCCACGATGCAGAAGC	GAAGGCGCTGTCTTGAGACC	NM_031545
**Col1A1**	CTGCACGAGTCACACCGGAA	CCAATGTCCAAGGGAGCCAC	NM_053304
**Col3A1**	TGGCTGCACTAAACACACTG	CCAATGTCATAGGGTGCGAT	NM_032085
**Rpl-32**	GGTGAAGCCCAAGATCGTCAA	TCTGGGTTTCCGCCAGTTTC	NM_013226.2
**Polr2a**	GGTATTGAGCAGATCAGCAAGG	CAATGCCCAGTACCGTGAAG	XM_343922

## Data Availability

Not applicable.

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
