# Peer review of "Sacubitril/Valsartan Improves Diastolic Function But Not Skeletal Muscle Function in a Rat Model of HFpEF"

_ijms, 2021, doi:10.3390/ijms22073570_

Round 1

Reviewer 1 Report

The research article "Sacubitril/Valsartan improves diastolic function but not skele3 tal muscle function in a rat model of HFpEF " analyzed the effects of sacubitril/valsartan  on the functionality of the myocardium, the skeletal muscle and the vasculature in a rat model of HFpEF.

The paper is well written and designed.  

In my opinion the article merit a direct acceptation for the following reasons

1.The experimental model of the study is correct and correspondent to the more frequent  clinical phenotype of HFpEF 

2. The results  of the paper have clearly a clinical significance.

3. In the discussion the existing literature is well analyzed and the clinical application of the results are well showed to the reader  

Reviewer 2 Report

Hypothesis: well understandable
Design: questionable
Methods: well organized
Results: clearly shown

The most important limitation of the study is lack of control animals treated by other anti-hypertensive drug than ARNI. As stated in page 14 3.2 Impact of Sac/Val on the myocardium lines 313-314, most of the alteration in myocardium may simply be due to pure blood pressure reduction. The statement before this part: "To estimate changes of the ventricular stiffness we determined LVEDP and the chamber stiffness constant βW, which are widely accepted to serve as load- and chamber size-independent parameters of passive chamber properties" is not the evidence of the statement: "it is unlikely that the observed reduced LV stiffness in the
314 treated group was a result of pure blood pressure reduction", because hypertension causes structural changes in myocardium. There are many results of ACE inhibitors, ARB, mineral corticoid receptor antagonist showing structual improvement of left ventricle but all these drugs have not shown to be effective for HFpEF. Show additional resutls which can distinguish the results of ARNI from that of ARB, MRA, or CCB (as shown in Croteau or Suematsu's results).

Minor points:

Page 12
2.8 Impact of Sac/Val on skeletal muscle atrophy and skeletal muscle function
Muslce weight/ toboa length ratio and CSA were reduced in obese than in lean animals. In addition, superiorEDL specific force was observed in lean rats than obese ones. Please explain the results. In humans, obese patients generally have more skeletal muscle mass and function probably due to higher resistance for skeletal muslce to counteract their heavier body weight. As a result, so-called "sarcopenic obesity" is observed in quite limited population of obese subjects.

Page 14
3.1 The female, obese ZSF1 rat as reseach model for HFpEF
ZSF1 rats showed significantly increased LVEDV than lean rats. It may not be a good model of HFpEF.

Page 15 lines 331-332
The statement "there was no evidence of heart failure and therefore the mouse served as a model of obesity induced cardiac remodeling rather than a model of HFpEF" is unclear unless stating what is the difference between obesity induced cardiac remodeling and a model of HFpEF.

Page 15 line 369
Show other site than tat of the mitral valve annulus, if possible. Otherwise readers can think the author takes only the place to support their hypothesis.

Pages 15-16 lines 373-381
Authors may be confused the significance of mitral valve function, trans-mitral flow, and tissue Doppler. E/e' is an indicator of diastole. However, mitral regurgitation is an indicator of systole.

Page 16 lines 398-401
"After 12 weeks of Sac/Val treatment we only measured a small improvement of the endothelial-dependent vasodilation of carotid arteries compared to the obese control group (Figure 7), suggesting that Sac/Val exerts its main effects on the myocardium and not on the vasculature"
A small improvement may lead to main effects if it is observed in systemic arteries (not limited to carotid arteries). So that it may be unknown the main effects of ARNI is on myocardium and not on the vasculature.

Round 2

Reviewer 2 Report

>The most important limitation of the study is lack of ...

The authors well clarify these points in the Comments and responses.
I understand that the structural change caused by hypertension could be reversed not only by ARNI but by other antihypertensive drugs, but it is out of scope of this study and lack of control animals is not a limitation of this study.

>Page 12 2.8 Impact of Sac/Val on skeletal muscle atrophy and skeletal muscle function / Muscle weight/ tibia length ratio and CSA were ...

If the CSA measurement in this study can distinguish intra muscular fat from skeletal muscle, it can explain reduced muscle weight/tibia length ratio and CSA and reduced EDL specific force in obese rats. If the CSA measurement in this study cannot distinguish intra muscular fat from skeletal muscle, it cannot explain them and it may be a limitation of this model. Please show how the CSA measurement was performed clearly.

>Page 14 3.1 The female, obese ZSF1 rat as research model for HFpEF / ZSF1 rats showed significantly increased LVEDV ...

Well answered.

>Page 15 lines 331-332 The statement "there was no evidence of heart failure and therefore the mouse ...

Well corrected.

>Page 15 line 369 Show other site than tat of the mitral valve annulus...

Please add the 2 reasons for that the authors focused on the septal mitral valve annulus to the method section.

>Pages 15-16 lines 373-381 Authors may be confused the significance of mitral valve function...

Well answered and corrected.

>Page 16 lines 398-401 "After 12 weeks of Sac/Val treatment we only measured a small improvement of the endothelial-dependent vasodilation of carotid arteries...

Well corrected.
